

# Antibiotic resistance and genomic features of *Clostridioides difficile* in southwest China

Wenpeng Gu[1], Wenge Li[2], Senquan Jia[1], Yongming Zhou[1], Jianwen Yin[1], Yuan Wu[2] and Xiaoqing Fu[1]

[1] Department of Acute Infectious Diseases Control and Prevention, Yunnan Provincial Centre for Disease Control and Prevention, Kunming, China
[2] State Key Laboratory of Infectious Disease Prevention and Control, National Institute for Communicable Disease Control and Prevention, Beijing, China

Corresponding author
Xiaoqing Fu, yncdcfxq@163.com

## ABSTRACT

**Background:** *Clostridioides difficile* infection (CDI) caused by toxigenic strains leads to antibiotic-related diarrhea, colitis, or even fatal pseudomembranous enteritis. Previously, we conducted a cross-sectional study on prevalence of CDI in southwest China. However, the antibiotics resistance and characteristics of genomes of these isolates are still unknown.

**Methods:** Antibiotic susceptibility testing with E-test strips and whole genome sequence analysis were used to characterize the features of these *C. difficile* isolates.

**Results:** Forty-nine strains of *C. difficile* were used in this study. Five isolates were non-toxigenic and the rest carried toxigenic genes. We have previously reported that ST35/RT046, ST3/RT001 and ST3/RT009 were the mostly distributed genotypes of strains in the children group. In this study, all the *C. difficile* isolates were sensitive to metronidazole, meropenem, amoxicillin/clavulanic acid and vancomycin. Most of the strains were resistant to erythromycin, gentamicin and clindamycin.

The annotated resistant genes, such as *macB*, *vanRA*, *vanRG*, *vanRM*, *arlR*, and *efrB* were mostly identified related to macrolide, glycopeptide, and fluoroquinolone resistance. Interestingly, 77.55% of the strains were considered as multi-drug resistant (MDR). Phylogenetic analysis based on core genome of bacteria revealed all the strains were divided into clade 1 and clade 4. The characteristics of genome diversity for clade 1 could be found. None of the isolates showed 18-bp deletion of *tcdC* as RT027 strain as described before, and polymorphism of *tcdB* showed a high degree of conservation than *tcdA* gene.

**Conclusions:** Most of the *C. difficile* isolates in this study were resistant to macrolide and aminoglycoside antibiotics. Moreover, the MDR strains were commonly found. All the isolates belonged to clade 1 and clade 4 according to phylogenetic analysis of bacterial genome, and highly genomic diversity of clade 1 was identified for these strains.

## INTRODUCTION

*Clostridioides difficile* is a Gram-positive and spore-producing anaerobic bacterium (*Abt, McKenney & Pamer, 2016*) Toxigenic *C. difficile* produce enterotoxin A (TcdA) and cytotoxin B (TcdB) leading to antibiotic-related diarrhea, colitis, or fatal pseudomembranous enteritis, collectively referred to as *C. difficile* infection (CDI) (*Janoir, 2016*; *Kuehne et al., 2010*). The bacterium can continue to circulate in humans, animals and the external environment. Patients are usually infected through the feco-oral route and the common clinical symptoms include diarrhea, to severe colitis, toxic megacolon, pseudomembranous enteritis and even intestinal perforation (*Vedantam et al., 2012*). The morbidity, mortality and disease burden of CDI are rising gradually, and it has become one of the major global public health problems (*Collins, Hawkey & Riley, 2013*). In recent years, with the widespread application of antibacterial drugs, the resistance of *C. difficile* has increased, and the emergences of multi-drug resistant strains have led to the increasing incidence and lethality of CDI (*Tang et al., 2016*).

Currently, some molecular typing methods have been applied to investigate the molecular epidemiology, genetic diversity and evolution of *C. difficile*. PCR ribo-typing (RT) and multi-locus sequence typing (MLST) are the major molecular typing methods for *C. difficile* at present, and these methods have similar distinguishing ability and are easy to share between different laboratories (*Collins, Elliott & Riley, 2015*; *Griffiths et al., 2010*). In the past 20 years, developments based on PCR and sequencing technologies, especially whole-genome sequencing, have greatly improved our understanding of the genetic diversity, epidemiology and pathogenicity of *C. difficile* (*He et al., 2010*). According to whole genomic sequencing, *C. difficile* is taxonomically divided into six phylogenetic clades, clade 1 to 5, and C-I (*Knight et al., 2015*). In China, most *C. difficile* strains belong to clade 1 with ST3, ST35 and ST54. In addition, the proportion of ST37 isolates from clade 4 is higher than other areas (*Xu et al., 2021*). The higher resistance rate to clindamycin, erythromycin and fluoroquinolones of clinical *C. difficile* strains are commonly found (*Wu et al., 2022*).

Previously, we conducted a cross-sectional study on prevalence and characteristics of CDI in southwest China (*Liao et al., 2018*). The results showed that ST35/RT046, ST54/RT012, ST3/RT001 and ST3/RT009 were the most distributed genotype profiles of isolated *C. difficile* strains. Among them, the major genotypes of strains in children patients were ST35/RT046, ST3/RT001 and ST3/RT009 compared with ST54/RT012 for adults. However, the antibiotic susceptibility and genomic features of these isolates were still not known. Knowledge regarding these fields maybe helpful to fully understand the characteristics of CDI in southwest China. This was the comprehensive study on *C. difficile* isolates in southwest China, and the results would be a helpful supplement and improvement to the national surveillance of *C. difficile* in China. The purpose of this study is deeply analyzing the *C. difficile* strains from our previous results by using antibiotic susceptibility tests and whole genomic sequencing method.

## MATERIALS AND METHODS

### Bacterial source, growth and DNA extraction

Forty-nine isolated *C. difficile* strains from our previous study (978 fecal samples) were used (Table S1). *C. difficile* were grown at 37 °C in an anaerobic chamber (Mitsubishi, Tokyo, Japan) by using an anaerobic gas pack (Thermo Fisher Scientific, Waltham, MA, USA). BHI agar (Oxoid, Basingstoke, Hampshire, United Kingdom) supplemented with 0.5% yeast extract (Oxoid, Basingstoke, Hampshire, United Kingdom) and 0.03% L-cysteine (Oxoid, Basingstoke, Hampshire, United Kingdom) was used for bacterial cultures, and plates were incubated for 24 to 48 h. Total genomic DNA of the isolated bacteria was extracted using a bacterial total genomic DNA extraction kit (Tiangen, Beijing, China) following the manufacturer's instructions. All DNA samples were stored at −20 °C for complete genome analysis.

### Antibiotic susceptibility tests

Susceptibility to metronidazole (MTZ), erythromycin (E), vancomycin (VA), meropenem (MEM), amoxicillin/clavulanic acid (AMC), ciprofloxacin (CIP), imipenem (IPM), tetracycline (TE), gentamicin (CN), clindamycin (DA), levofloxacin (LEV), ceftazidime (CAZ), amoxicillin (AML), oxacillin (OX) and cefotaxime (CTX) were performed using E-test strips (Thermo Fisher Scientific, Waltham, MA, USA) (*Li et al., 2019*). The strains were adjusted with 0.85% saline to 1 McFarland standard and swabbed onto Mueller-Hinton (MH) agar supplemented with 5% horse blood, heme (5 μg/mL) and Vitamin K1 (1 μg/mL). Plates were incubated anaerobically at 37 °C and MIC breakpoints were read after 48 h.

Breakpoints for these antibiotics were determined according to the recommendations of the Clinical and Laboratory Standards Institute (CLSI) M11-A7 and M100-S24. *C. difficile* ATCC700057 was used as a quality control. Multi-drug resistance (MDR) was defined as acquired non-susceptibility to at least one agent in three or more antimicrobial categories (*Li et al., 2019*; *Magiorakos et al., 2012*).

### Genome sequencing, assembly and annotation

Genomic sequencing of all isolates was performed by our laboratory on the Illumina MiSeq platform using $2 \times 150$ bp paired-end reads (*Gu et al., 2020*). The libraries were built by using a Nextera XT DNA Library Prep Kit following the manufacturer's instructions. In general, 1 ng genomic DNA of each strain was used. The segment and purification were performed, then index PCR was used to add the barcodes using i5 and i7 primers, after that the purification was executed again. Finally, the libraries were normalized, pooled and sequenced using an Illumina MiSeq sequencing system (Illumina, San Diego, CA, USA).

The raw data were trimmed and filtered by Trimmomatic (version 0.38) (*Bolger, Lohse & Usadel, 2014*). SOAP denovo (version 2.04) was used to assemble the draft genomes with k-mer values optimized to the best assembly results (*Li et al., 2009*). The open reading frame (ORF) of each genome was predicted by GenemarkS software (version 4.28) (*Besemer, Lomsadze & Borodovsky, 2001*) with default parameters. The predicted amino acid sequences were aligned and annotated by DIAMOND (E-value: 1e−5, top 5)

(*Buchfink, Xie & Huson, 2015*) to SwissProt, NR, COG, KEGG, Virulence Factor Database (VFDB), Antibiotic Resistance Genes Database (CARD), and Pathogen-Host Interaction database (PHI). The heatmaps based on CARD and VFDB database annotations were drawn using pheatmap in the R package (version 3.2). CD630 with accession number: CP010905 was used as the reference strain for alignment.

## Phylogenetic analysis and statistics

Phylogenetic analyses based on whole genomes were performed for all 49 *C. difficile* in this study with 20 reference strains isolated from different countries in different years (Table S2). The core SNP alignment phylogenetic tree was built based on the SNP of concatenated core genes with Snippy (*Bush, 2021*), and the recombination region removed with BAGEP (*Olawoye, Frost & Happi, 2020*).

*tcdA*, *tcdB* and *tcdC* genes were extracted by using CLC genomic workbench, and only full length of these genes were used for alignment. The sequences of each strain were aligned by MEGA 6.0 using the neighbour-joining (NJ) method with 1,000 bootstrap replicates to generate a phylogenetic tree. CD630 (accession number: CP010905) and R20291 (accession number: FN545816) were used as references.

Statistical analysis was performed using the SPSS software package (version 16.0, IBM, Armonk, NY, USA). Kolmogorov-Smirnov Z, T-test or $\chi$2 test were used where appropriate. A *P* value of <0.05 was recognized as statistically significant.

## Availability of data and materials

All data generated or analysed during this study are included in this published article. The sequence data have been deposited into the National Center for Biotechnology Information (NCBI), https://www.ncbi.nlm.nih.gov/ with BioProject accession number: PRJNA785426 (SRR17146436–SRR17146484).

## Ethics statements

The human sample collection and detection protocols were carried out in accordance with relevant guidelines and regulations. All experimental procedures were approved by the Ethics Review Committee (Institutional Review Board (IRB)) of Yunnan Provincial Centre for Disease Control and Prevention (Number: YNCDC-2017005). All adult subjects provided written consent, and a parent or guardian of any child participant provided written consent on their behalf.

# RESULTS

## Antibiotic resistance

All the strains were sensitive to MTZ, MEM, AMC and VA. All of the isolates were intermediately resistant to CTX. However, 87.80% (43/49) of the *C. difficile* strains were resistant to erythromycin, 81.60% (40/49) to CN, 61.20% (30/49) to DA, 36.70% (18/49) to CAZ, and 28.60% (14/49) of isolates to LEV (Table 1). Among all the *C. difficile* isolates in this study, 77.55% (38/49) were defined as MDR strains. Twenty-four strains were

**Table 1 Comparison of antibiotic resistance between children and adults in this study.**

| Antibiotics | Sensitive | | | | Intermediate | | | | Resistant | | | | χ2 | P value |
|---|---|---|---|---|---|---|---|---|---|---|---|---|---|---|
| | Children | | Adults | | Children | | Adults | | Children | | Adults | | | |
| MTZ | 40 | 81.63% | 9 | 18.37% | 0 | – | 0 | – | 0 | – | 0 | – | – | – |
| E | 5 | 10.20% | 1 | 2.04% | 0 | – | 0 | – | 35 | 71.43% | 8 | 16.33% | 0.013 | 0.909 |
| MEM | 40 | 81.63% | 9 | 18.37% | 0 | – | 0 | – | 0 | – | 0 | – | – | – |
| AMC | 40 | 81.63% | 9 | 18.37% | 0 | – | 0 | – | 0 | – | 0 | – | – | – |
| CIP | 18 | 36.73% | 4 | 8.16% | 21 | 42.86% | 4 | 8.16% | 1 | 2.04% | 1 | 2.05% | 1.429 | 0.490 |
| IPM | 38 | 77.55% | 9 | 18.37% | 1 | 2.04% | 0 | – | 1 | 2.04% | 0 | – | 0.469 | 0.791 |
| TE | 37 | 75.51% | 8 | 16.33% | 2 | 4.08% | 1 | 2.04% | 1 | 2.04% | 0 | – | 0.684 | 0.710 |
| CN | 0 | – | 2 | 4.08% | 6 | 12.24% | 1 | 2.04% | 34 | 69.39% | 6 | 12.25% | 9.269 | 0.010 |
| DA | 9 | 18.37% | 4 | 8.16% | 6 | 12.24% | 0 | – | 25 | 51.02% | 5 | 10.21% | 2.741 | 0.254 |
| LEV | 7 | 14.29% | 2 | 4.08% | 23 | 46.94% | 3 | 6.12% | 10 | 20.41% | 4 | 8.16% | 1.870 | 0.393 |
| VA | 40 | 81.63% | 9 | 18.37% | 0 | – | 0 | – | 0 | – | 0 | – | – | – |
| CAZ | 9 | 18.37% | 3 | 6.12% | 16 | 32.65% | 3 | 6.12% | 15 | 30.61% | 3 | 6.13% | 0.471 | 0.790 |
| AML | 36 | 73.47% | 9 | 18.37% | 4 | 8.16% | 0 | – | 0 | – | 0 | – | 0.980 | 0.322 |
| OX | 35 | 71.43% | 9 | 18.37% | 3 | 6.12% | 0 | – | 2 | 4.08% | 0 | – | 1.253 | 0.535 |
| CTX | 0 | – | 0 | – | 40 | 81.63% | 9 | 18.37% | 0 | – | 0 | – | – | – |

Note:
Breakpoints of MIC cutoff values for each antibiotic. MTZ (metronidazole): sensitive (S): 8 μg/ml; intermediately resistant (I) 16 μg/ml; resistant (R) 32 μg/ml. E (erythromycin): S: ≤ 0.25 μg/ml; I: 0.5 μg/ml; R ≥ 1 μg/ml. MEM (meropenem): S: 4 μg/ml; I: 8 μg/ml; R: 16 μg/ml. AMC (amoxicillin/clavulanic acid): S: 4/2 μg/ml; I: 8/4 μg/ml; R: 16/8 μg/ml. CIP (ciprofloxacin): S: 2 μg/ml; I: 4 μg/ml; R: 8 μg/ml. IPM (imipenem): S: 4 μg/ml; I: 8 μg/ml; R: 16 μg/ml. TE (tetracycline): S: 4 μg/ml; I: 8 μg/ml; R: 16 μg/ml. CN (gentamicin): S: 2 μg/ml; I: 4 μg/ml; R: 8 μg/ml. DA (clindamycin): S: 2 μg/ml; I: 4 μg/ml; R: 8 μg/ml. LEV (levofloxacin): S: 2 μg/ml; I: 4 μg/ml; R: 8 μg/ml. VA (vancomycin): S: 4 μg/ml; I: 8 μg/ml; R: 32 μg/ml. CAZ (ceftazidime): S: 16 μg/ml; I: 32 μg/ml; R: 64 μg/ml. AML (amoxicillin): S: 0.5 μg/ml; I: 1 μg/ml; R: 2 μg/ml. OX (oxacillin): S: 32 μg/ml; I: 64 μg/ml; R: 128 μg/ml. CTX (cefotaxime): S: 16 μg/ml; I: 32 μg/ml; R: 64 μg/ml. We used the chi-square test of the row × column of multiple rates or multiple constituent ratios in SPSS software.

resistant to three classes of antibiotics, nine strains were resistant to four classes, three isolates were resistant to five classes, and two *C. difficile* isolates resistant to six classes.

We further compared the antibiotic resistance results between children and adult groups (Table 1). Except the gentamicin, there was no statistical significance between two groups for all antibiotics. The resistant rate of gentamicin in children group (69.39%) was significantly higher than adult group (12.25%) (χ2 = 9.269, P = 0.010).

## Genomic sequencing results

The general information and details of genomic sequencing results are shown in Table 2 and Table S3. The effect reads after quality control of raw data were 91.39 ± 2.02 (%) for all the 49 *C. difficile* strains in this study, and the average nucleotide identity (ANI) was 99.65 ± 0.48. The average genome size was 4.07 ± 0.10 Mbp, GC content was 28.86 ± 0.35 (%), and the coverage for all strains was 98.14 ± 0.68 (%). The reference CD630 strain showed 4.27 Mbp genome size and 29.00% GC content. For variation analysis, the average SNPs of all the *C. difficile* strains was 16,665.84 ± 9,676.51 bp, from 3,023 bp to 36,394 bp. There was 116.69 ± 59.12 bp insertion, and 144.39 ± 75.04 bp deletion of all the bacteria.

**Table 2 The genomic sequencing results of *C. difficile* in this study.**

|  | Mean ± STD (*n* = 49) | Min | Max |
|---|---|---|---|
| Effect reads (%) | 91.39 ± 2.02 | 86 | 94 |
| ANI | 99.65 ± 0.48 | 99 | 100 |
| Completeness (%) | 98.33 ± 1.21 | 94 | 100 |
| Contamination (%) | 1.04 ± 0.54 | 0 | 2 |
| Genome size (Kbp) | 4,065.39 ± 103.80 | 3,767 | 4,210 |
| N50 (kbp) | 11.00 ± 5.44 | 3 | 27 |
| GC (%) | 28.86 ± 0.35 | 28 | 29 |
| Coverage (%) | 98.14 ± 0.68 | 97 | 99 |
| SNPs | 16,665.84 ± 9,676.51 | 3,023 | 36,394 |
| Ins | 116.69 ± 59.12 | 31 | 234 |
| Del | 144.39 ± 75.04 | 39 | 275 |

**Note:**
ANI, Average Nucleotide Identity.

## Annotation results

The annotation results of *C. difficile* isolates in this study revealed that the average numbers of genes annotated using the different databases were 3,925.27 ± 129.82. There were 3,873.78 ± 120.07, 1,938.49 ± 33.77, 2,949.37 ± 43.02, and 1,236.92 ± 8.26 genes annotated with NR, KEGG, COG and SwissProt databases, respectively, as shown in Fig. 1A. Furthermore, 132.47 ± 2.14, 93.84 ± 4.86, and 193.63 ± 3.30 genes were annotated using PHI, CARD and VFDB databases (Fig. 1A). Specifically, *macB* (19.78%) belonged to ATP-binding cassette (ABC) antibiotic efflux pump resistant to macrolide antibiotic was the most distributed gene in CARD database. *arlR* (4.98%) and *efrB* (4.32%) were also the top distributed genes in CARD database, both belonged to antibiotic efflux pump. *vanRA* (5.62%), *vanRG* (9.45%), and *vanRM* (3.90%) belonged to glycopeptide resistance gene cluster were the top distributed genes for glycopeptide antibiotic resistance, as Fig. 1B shown. For VFDB annotation, two clusters of strains were generated according to the virulence genes distribution (Fig. 1C). From YNCD947 to YNCD55 isolates, secretion system related genes were highly distributed, however from YNCD6 to YNCD505 strains, the toxin related genes were highly distributed.

## Phylogenetic analysis

Phylogenetic analysis was conducted on the 49 *C. difficile* isolates used in this study with core genomes of bacteria, as shown in Fig. 2. Five clusters of branches were obviously identified, and each cluster belonged to a different clade, namely clade 1 to 5, as shown in Fig. 2. All the strains in this study were divided into clade 1 and clade 4. Except three isolates that belonged to clade 4 (ST39 strains; YNCD887, YNCD175 and YNCD372), all others belonged to clade 1. However, there were some sub-clusters found in clade 1, and each sub-cluster corresponded to the major sequence type of the strains. For example, from YNCD905 to YNCD947, all these strains belonged to ST35; from YNCD601 to YNCD916, all belonged to ST3; from YNCD086 to YNCD088, both belonged to ST2; from
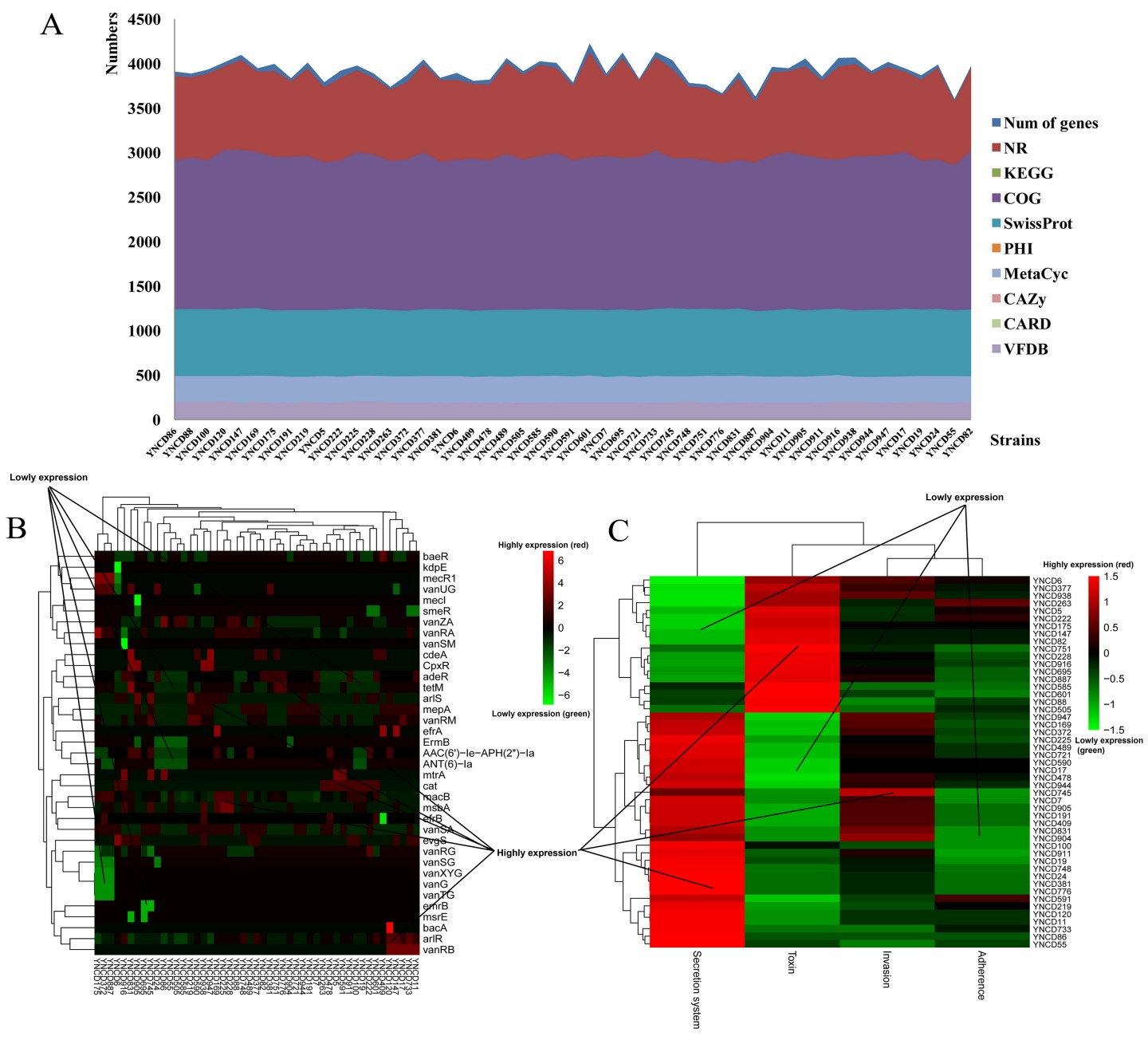

**Figure 1 The annotation results of *C. difficile* strains in this study.** (A) The numbers of genes annotated with different database. (B) The heatmap of antibiotic resistant genes annotated using the Comprehensive Antibiotic Resistance database (CARD). (C) The heatmap of virulence genes annotated using the virulence factor database (VFDB).

YNCD082 to YNCD147, all belonged to ST54 (Fig. 2). The high genomic diversity characteristic of clade 1 could be detected in the strains that were grouped into this clade.

## Toxin gene analysis

Toxin gene analysis revealed the presence of the full length *tcdA* gene in 18 *C. difficile* strains studied, and the alignment results indicated that six clusters were generated

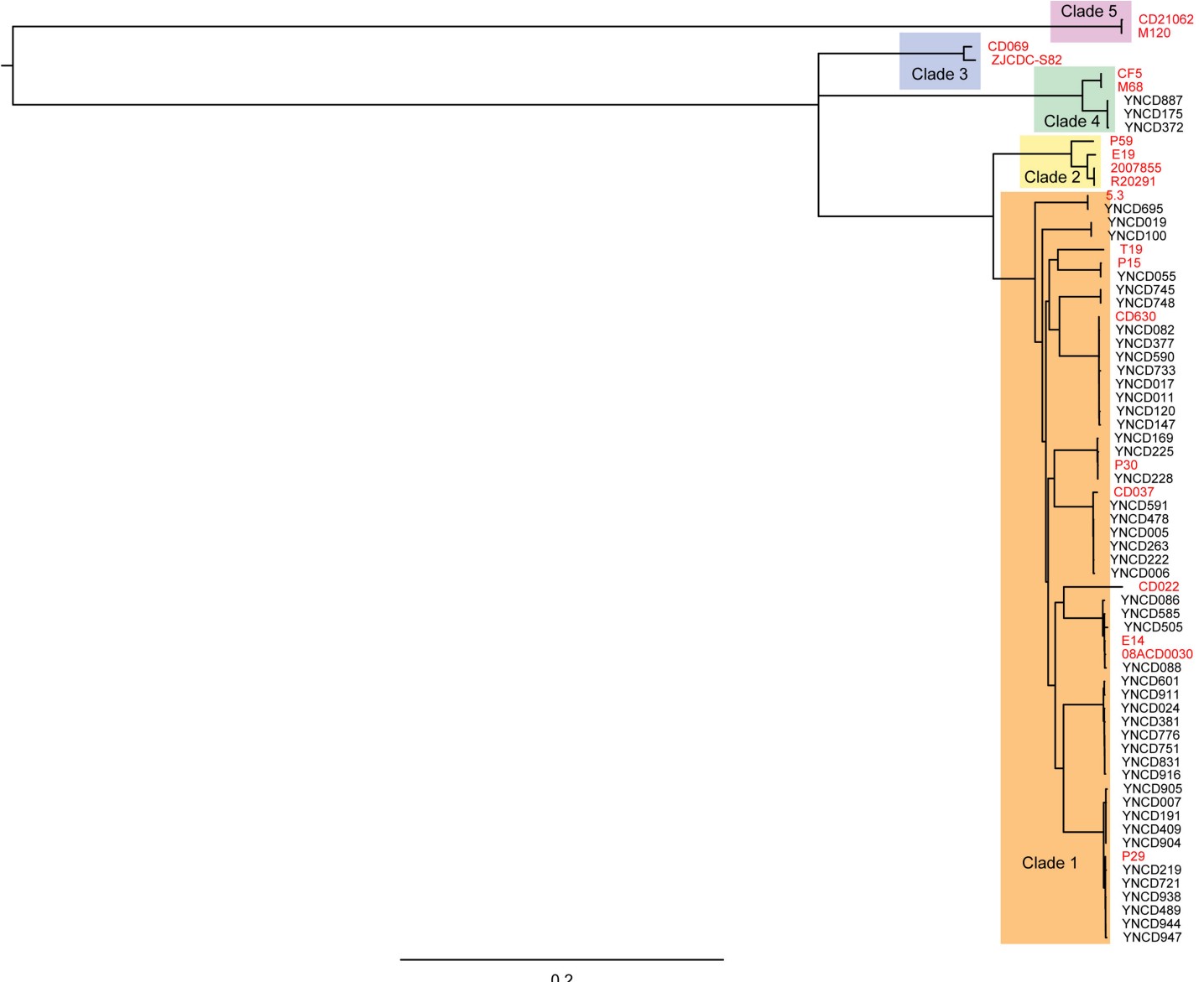

**Figure 2 Phylogenetic tree based on core genomes of *C. difficile* isolates in this study.** The reference strains were marked in red font, and the usage of different colors for different clades.

according to the gene polymorphism, as shown in Fig. 3A. Compared with highly pathogenic strain (R20291, RT027), all the *tcdA* genes were similar with low pathogenic strain (CD630, RT012). For *tcdB* gene, 28 full lengths of this gene were extracted, but only four clusters were generated, which showed a high degree of conservation (Fig. 3B). Similar with *tcdA*, all the *tcdB* genes were clustered with low pathogenic strain CD630, but far away from highly pathogenic strain R20291. The *tcdC* analysis revealed that 17 full lengths of genes were extracted, and all the strains did not have the 18-bp deletion of R20291 from 391 to 408 nt of the gene, as shown in Fig. 3C. Only some

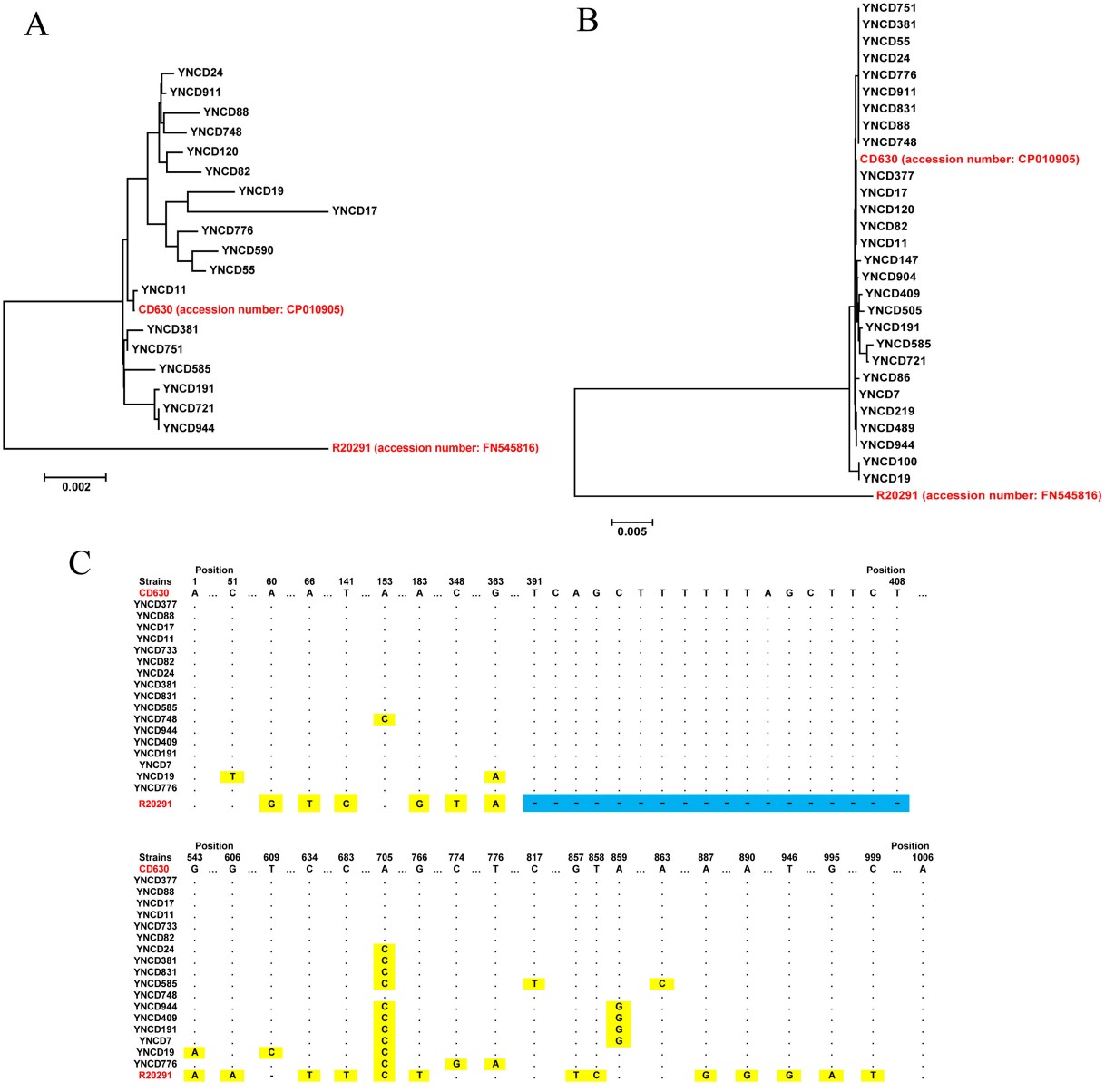

**Figure 3 The toxin gene analysis of all *C. difficile* isolates in this study.** (A) The phylogenetic tree of the tcdA gene. (B) The phylogenetic tree of the tcdB gene. (C) The tcdC gene polymorphism analysis of the strains; blue area indicates 18-bp deletion of the gene, and yellow area indicates point mutations of the gene. The red font indicates reference strains.

of *C. difficile* isolates in this study had point mutations although they were all similar to CD630.

## DISCUSSION

Antibiotic usage has been associated with CDI in clinics, which disrupt the normal gut microbiota of hosts and lead to the proliferation of *C. difficile*, especially for macrolides, aminoglycosides, quinolones and cephalosporins. Previous studies revealed that clade 4 of *C. difficile* often associated with MDR, and was recognized as primary clade of MDR after clade 1 in China (*Jin et al., 2017*; *Li et al., 2019*). Several European countries have

reported the reduced metronidazole susceptibility of strains, but still not found in China (*Peng et al., 2017*; *Spigaglia, Barbanti & Mastrantonio, 2011*). In our study, all the strains were sensitive to metronidazole, meropenem, amoxicillin/clavulanic acid and vancomycin, which showed identical results as previously reported. *Li et al. (2019)* performed a study on antibiotic resistance in *C. difficile* isolated from clinical sources across China. In general, 319 isolates tested by using 11 antibiotics, 313 were resistant to at least one antibiotic. The ciprofloxacin, clindamycin, and erythromycin were the highest rate of resistant antibiotics among all age groups. All isolates were susceptible to metronidazole and vancomycin. Moreover, the proportion of multi-drug resistant strains was much lower than previous reports, except for chloramphenicol and meropenem. In this study, most of the strains were resistant to erythromycin, gentamicin and clindamycin, followed by ceftazidime and levofloxacin. The CARD annotation results were consistent to phenotypic results, *macB*, *vanRA*, *vanRG*, *vanRM*, *arlR*, and *efrB* were resistant genes referred to macrolide, glycopeptide, and fluoroquinolone. There were some associations between the phenotypic resistance and the annotated antibiotic resistance genes, except for *vanRA*, *vanRG* and *vanRM* and glycopeptide resistance, which was not phenotypically tested. The *macB*, *arlR* and *efrB* genes were all belonged to antibiotic efflux pump for macrolide and fluoroquinolone resistance, in accordance with the erythromycin and levofloxacin resistant phenotypes. Interestingly, the MDR strains showed extremely high proportion than other areas of China, and 77.55% were defined as MDR.

Clade 1 represents highly heterogeneous toxigenic and non-toxigenic sequence types and ribo-types, including many clinically significant strains, such as RT014, RT002 and RT018, some of the most commonly isolated types of RTs from CDI patients (*Bauer et al., 2011*; *Collins, Hawkey & Riley, 2013*; *Foster et al., 2014*; *Lim et al., 2014*). Clade 4 contains RT017 (ST37), which has a different toxin spectrum and is generally resistant to clindamycin and fluoroquinolone. Despite the absence of toxin A and CDT toxin expression, RT017 still causes widespread CDI, which is associated with outbreaks in Europe and North America (*Alfa et al., 2000*; *Drudy et al., 2007*; *Goorhuis et al., 2009*). In addition, RT017 is responsible for the most cases of CDI in Asia (*Li et al., 2019*; *Wu et al., 2022*). Clade 2 contains highly toxic RT027 and other clinically important RTs, including RT244 and RT176 (*Valiente et al., 2012*). Recent MLST and WGS studies have shown that the heterogeneity of clade 5 is more diverse than originally imagined, including not only RT078, but also RTs from numerous clinical, animal and food sources (RT033, RT045, RT066, RT126, RT127, RT237, RT280, RT281 and RT288 strains) (*Knight et al., 2019*; *Wu et al., 2019*). In this study, only clade 1 and 4 of strains were identified, and isolates in clade 1 revealed characteristics of highly genomic diversity. The sub-clusters of strains were consistency with ST types in clade 1.

Both *tcdA* and *tcdB* are located at the 19.6 kb pathogenicity locus (PaLoc). *tcdC* is located downstream of *tcdA* and is transcribed from the opposite direction of the two toxin genes (*Pruitt et al., 2010*). The decrease in TcdC expression corresponds to the increase in TcdA and TcdB expression, indicating that TcdC may be a negative regulator of toxin production (*Matamouros, England & Dupuy, 2007*). However, the role of *tcdC* in

repression of PaLoc toxin expression has been subsequently disproven (*Bakker et al., 2012*). RT027 (highly pathogenic clone) strain from the United Kingdom showed 18-bp deletion for *tcdC* gene, and this deletion indicated to be conserved among all RT027 isolates (*Curry et al., 2007*). Furthermore, previous report demonstrated that epidemic *C. difficile* that produced high levels of toxins carried deletion or frame shift mutations in *tcdC* gene (*Curry et al., 2007*). In this study, none of the isolates showed 18-bp deletion as RT027 strain as described before, and it was considered that all these *C. difficile* were low pathogenic clones. Moreover, the *tcdB* gene was more conserved than the *tcdA* gene. *Steele et al. (2013)* investigated the efficacy of specific human monoclonal antibodies against TcdA and TcdB toxins. Their results showed that TcdB was more important than TcdA as virulence factor associated with gastrointestinal tract disease. Therefore, the nucleotide polymorphisms probably were associated with pathogenic ability of these toxins.

## CONCLUSIONS

In this study, the antibiotic susceptibility tests and whole genomic sequencing method were both used to analyze the features of *C. difficile* in southwest China. Most of the strains were resistant to erythromycin, gentamicin and clindamycin. A total of 77.55% of the strains were considered as MDR. Phylogenetic analysis based on core genome of bacteria revealed all the strains were divided into clade 1 and clade 4. The characteristics of genome diversity for clade 1 could be found. None of the isolates showed 18-bp deletion of *tcdC* as RT027 strain as described before, and the *tcdB* gene was more conserved than the *tcdA* gene.

### Funding
The authors received no funding for this work.

### Competing Interests
The authors declare that they have no competing interests.

### Author Contributions
- Wenpeng Gu performed the experiments, prepared figures and/or tables, authored or reviewed drafts of the article, and approved the final draft.
- Wenge Li performed the experiments, prepared figures and/or tables, and approved the final draft.
- Senquan Jia performed the experiments, prepared figures and/or tables, and approved the final draft.
- Yongming Zhou analyzed the data, prepared figures and/or tables, and approved the final draft.
- Jianwen Yin analyzed the data, prepared figures and/or tables, and approved the final draft.
- Yuan Wu conceived and designed the experiments, authored or reviewed drafts of the article, and approved the final draft.

- Xiaoqing Fu conceived and designed the experiments, authored or reviewed drafts of the article, and approved the final draft.

## Human Ethics

The following information was supplied relating to ethical approvals (*i.e.*, approving body and any reference numbers):

The human sample collection and detection protocols were carried out in accordance with relevant guidelines and regulations. All experimental procedures were approved by the Ethics Review Committee [Institutional Review Board (IRB)] of Yunnan Provincial Centre for Disease Control and Prevention (Number: YNCDC-2017005).

## Data Availability

The sequence data are available at the National Center for Biotechnology Information (NCBI): PRJNA785426.

## Supplemental Information

Supplemental information for this article can be found online at http://dx.doi.org/10.7717/peerj.14016#supplemental-information.

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
