# Peer review of "Antibiotic resistance and genomic features of Clostridioides difficile in southwest China"

_PeerJ, doi:10.7717/peerj.14016_

## Round 0.1 · original submission · Major Revisions

Please revise the manuscript according to reviewers' comments, especially on writing, address the questions and concerns of the reviewers.

·

Basic reporting

This study described by Gu WP, et al presented antibiotic resistance and genomic characteristics of C.difficile isolated from Southwest China. The English writing is good and the statement is clear. The authors should add more references related CDI in China. The results including figures and tables are enough to present the main findings in this study. However, More works are needed in order to deeply improve the quality of this study.
1. The abstract should be re-written, the version is not acceptable. The conclusions are not reasonable.
2. In Introduction, authors should focus on the current CDI in China including molecular epidemiology, antibiotic resistance, and etc.
3. The description of genome sequencing, assembly and annotation is not clear. Which genome did you used as a reference for alignment?
4. In Results, authors presented results of some genes, but the methods are not mentioned.
5. Due to the small number of isolates, the comparative analysis among different groups on antibiotic resistance is not meaningful.
6. Through phylogenetic analysis, the results are very confused. Authors should mainly state or present the findings in this study.
7. The discussion need to be re-written based on the study.

No comment.

Experimental design

The methods should be revised, more information need to be provided.


No comment.

Validity of the findings

The authors should carefully analyzed the data and make a solid conclusion.


No comment.

Additional comments

N/A

Reviewer 2 ·

Basic reporting

This manuscript entitled “Antibiotic resistance and genomic features of Clostridioides difficile in Southwest China” aimed to characterize the antibiotic resistance and genomic features of C. difficile isolates previously reported by the authors from Southwest China.
While the study shows some interesting findings, some major and minor issues enumerated below need to be addressed before it may be considered for publication. Also, the language needs to be edited and largely improved overall by a person proficient in English for grammatical accuracy and clarity. (Some suggestions for language improvement have been given below). Poor language usage may have unfortunately distorted the meaning intended by the authors and has impaired the readability in numerous places of the manuscript.

Major issues/ suggestions/ comments:

1. Lines 118-119: The sentence “ST35/RT046, ST3/RT001 and ST3/RT009 were the commonly found in children patients but ST54/RT012 for adults” has been copied word-to-word from the abstract of the authors’ previous publication in Scientific Reports (DOI:10.1038/s41598-018-21762-7). This needs to be re-worded to avoid plagiarism and for grammatical accuracy.

2. Lines 119-120: The sentence “However, the antibiotics resistance and characteristics of genomes for these isolates were still unknown” has been repeated word-to-word from the Abstract (lines 50-51). Please rephrase this.

3. Lines 122-123: Again, this has been repeated word-to-word from the Abstract (lines 53-54). Please rephrase.

4. Line 296-321: Prevalence and the age-related demographic distribution of CDI in the region under study, ribotyping, MLST and profiling of toxin genes have been described, discussed and published in the authors’ previous publication in Scientific Reports and is not the focus of this manuscript. Unless discussed in reference to the previous publication on the findings relevant to the focus of this manuscript, this section needs to be removed.


Minor issues/ suggestions/ comments:

Abstract:

• Line 1: “lead” needs to be changed to “leads” for grammatical accuracy.

• Remove the word ‘severely’ preceding ‘fatal’ (line 48 in Abstract and line 92 in Introduction), as fatality has no range of severity.

• Line 50: “for”. Suggested edit: “of”.

• Lines 53-54: “In this study, the antibiotic susceptibility tests and whole genomic sequencing method were both used to further analyze the features of C. difficile”. Suggested edit: “Antibiotic susceptibility testing with Etest strips and whole genome sequence analysis were used to characterize the features of these C. difficile isolates”.

• Line 56: “The results showed 49 strains of C. difficile were involved” – The meaning is not clear.

• Line 57 : “rests” should be changed to “rest”.

• Lines 58-59: “All the C. difficile”. Suggested change: : “All the C. difficile isolates”.

• Line 62: “were mostly identified referred to” – meaning is not clear.

• Line 64: “resistance” needs to be changed to “resistant” for grammatical accuracy.

• Lines 68, 287: “conservatism” should be changed to “conservation” for scientific accuracy.

• The Conclusion should clarify which features were demonstrated rather than merely stating that features were demonstrated.


Introduction:


• Line 94: What is the basis for saying that C. difficile has a unique life cycle? There are many bacteria that circulate among humans, animals and external environment, therefore this is not a unique feature.

• Line 96: “fecal or oral route” needs to be changed to “feco-oral route” for accuracy.

• Line 97: The mild carrier state is not a clinical symptom.

• “C. difficile” needs to be altered to either “C. difficile isolates” or “C. difficile strains” as appropriate for accuracy in lines such as 118, 128, 222, 225, 228, 236, 241, 247, 267, 292, 537 and Figure 3 and 4 legends.

• Lines 121-122: A brief explanation on the importance of ‘fully understand(ing) the features of CDI in southwest China’ will be of interest to the readers here.


Methods:

• Line 137: Remove the word “resistance” preceding “susceptibility testing” for accuracy.

• Line 143: “McFand” should be corrected to “McFarland standard and…”.

• Line 150: I suggest removing the word “strain” after (MDR) for grammatical accuracy.

• Line 150: Definition of MDR “resistance to at least three antibiotic classes” does not clarify if an organism needs to be resistant to all antibiotics of the antibiotic classes under consideration to be considered MDR. More clarity could be obtained by using the definition used by Magiorakos et al (2012; DOI: 10.1111/j.1469-0691.2011.03570.x) referring to at least one agent in three or more antimicrobial categories.

• Line 176: Re-phrase “reference strains from different years and countries” for accuracy. Suggestion – “reference strains isolated from different countries in different years”.

• Line 179: What is meant by “gubbins” is unclear. Please revise.

• Lines 185-186: Inclusion of the range of accession numbers of the sequences deposited in the data deposition statement will make data accession easier for the readers.

• Line 188: “if appropriate”. Suggested change – “where appropriate”.

• Lines 185-186 and 191-195: The same information has been repeated. Lines 191-195 can be removed.

• The submitted human participant information file needs to be provided in English.


Results:

• Line 222: Correct “intermediate” to “intermediately resistant” for grammatical accuracy.

• Lines 223-224: The phrase “were resistant” can be removed after the first mention to avoid language redundancy.

• Line 224: Include the proportion of isolates (14/49) resistant to LEV for uniformity.

• Line 226: The phrase “In general,” can be removed.

• Line 230: Correct “group” to “groups”.

• Table 1: Suggested edit for title – “Comparison of antibiotic resistance between children and adults in this study”

• Table 1: MIC cutoff vales/ ranges of values used for determining isolates as sensitive, intermediately resistant or resistant and the reference source/s of those breakpoints (e.g. CLSI) need to be included in this Table.

• Full names of the antibiotic abbreviations used need to be mentioned as a footnote in the Table.

• Table 2: Mention what is meant by abbreviation the “ANI” in a footnote.

• Table 2: Mention the number of strains tested (n=49).

• Change “were” to “are” line 235 and “was” to “were” in line 236.

• Mention of Table 2 in line 235 and Supplementary Table 3 in lines 243-244 can be combined into one sentence.

• Lines 234-244: The “Genome sequencing results” section can be made more concise. A brief description/ mention of how measures such as genome size and GC content relate to previously reported C. difficile strains is needed.

• Lines 247-248: “the average numbers of genes were 3925.27±129.82” – Do you mean ‘the average numbers of genes annotated using the different databases used in this study’?

• Lines 248-250: Suggestion for language correction – “There were 3873.78±120.07, 1938.49±33.77, 2949.37±43.02, and 1236.92±8.26 genes annotated with NR, KEGG, COG and SwissProt databases, respectively, as shown in Figure 2A.”

• Line 251: “annotated to PHI, CARD and VFDB”. Suggested language edit – “annotated using PHI, CARD and VFDB databases”.

• Figure 2 legend: Spell out the abbreviations CARD and VFDB.

• Figure 2 legend: Suggested language edit - Replace “in” with either “using” or with.

• Lines 252-261: It will be clearer if the detected antibiotic resistance genes are organized according to some criteria such as mechanism of antimicrobial action or drug classes (e.g. different efflux pump genes described one after the other).

• Lines 252-264: Please edit and largely improve the language in this section for accuracy of your message. The points I assume you are trying to make are being distorted because of poor language use.

• Line 257-260: “efrB (4.32%) was the fluoroquinolone antibiotic, macrolide antibiotic and rifamycin antibiotic resistant gene belonged to ATP-binding cassette (ABC) antibiotic efflux pump” – Makes no sense.

• Lines 267-268: Suggested language edit – “Phylogenetic analysis was conducted on the 49 C. difficile isolates used in this study with…”

• Line 269, 271, 284, 291: “As Figure X shown”. Language edit – “As shown in Figure X”.

• Lines 268-269 – The statement that reference strains were marked in red font in the phylogenetic tree is better moved to the Figure legends of Figures 3 and 4.

• Line 271: The mention of the usage of different colors for different clades is better moved to the Figure legend.

• Suggested language edits – “belonged to a different clade” (line 270), “isolates that belonged to” (line 272), “all others” (line 273), “each sub-cluster corresponded to the major sequence type of the strains” (lines 274-275), “Toxin gene analysis” (line 281, Figure 4 title), “Antibiotic usage has been associated” (line 322), “Previous studies” (lines 324-325).

• Line 282: “Eighteen C. difficile strains for full lengths of tcdA gene were extracted” – Do you mean ‘Toxin gene analysis revealed the presence of the full length tcdA gene in 18 C. difficile strains studied’?

• Figure 4 legend: Suggested language edits - Change “indicated” to “indicates”, “analyses among all the strains” to “analysis of the strains” and “point mutation” to “point mutations”.

• Figure 4 legend: The phrase “red font indicates reference strains” need not be repeated and can be moved to the end of the legend.

• Line 290: Edit “didn’t had”. Suggestion – “did not have…”

• Lines 291-293: “All the tcdC genes of C. difficile in this study were similar with CD630 presented some point mutation” – The meaning is not clear. Do you mean that all the tcdC-containing strains had a point mutation/s compared to CD630 or that only some of them did although they were all similar to CD630? (Figure 4C supports the latter).


Discussion:

• Lines 322-387: Please improve language for readability and accuracy.

• Line 335-336: “the proportion of multi-drug resistant strains was much lower than previous reports” – Except for chloramphenicol meropenem. Especially the latter is better mentioned, as meropenem was used in the present study.

• Lines 342-344: Please remove the sentence “In general, 24 strains were resistant to three classes of antibiotics, 9 strains were resistant to four classes, 3 isolates were resistant to five classes, and 2 C. difficile resistant to six classes” that has been repeated ad verbatim from the Results section.

• Line 345: Please edit “toxic and non-toxic” as “toxigenic and non-toxigenic” for scientific accuracy.

• Lines 345-361: The description of the clades need to be structured to better focus on the clades detected in the present study.

• Lines 352-353: “RT017 still causes widespread CDI, which is associated with outbreaks in Europe and North America” – It is also important to highlight that RT017 is responsible for the most cases of CDI in Asia, which is highly relevant to the present study (Li et al 2019 DOI: 10.1016/j.anaerobe.2019.102094).

• Line 374: Alter “conservative” to “conserved” for scientific accuracy.


Conclusions:

• The conclusions section should be adjusted based on the points discussed in other sections.

Experimental design

1. The authors have conducted antibiotic susceptibility testing and genome analysis of 49 C. difficile isolates selected from 978 isolates they have previously reported from Southwest China. However, the basis for the selection of these 49 isolates from the previously reported 978 isolates has not explained. This explanation may affect the validity and the importance of many of the findings presented in this study, and therefore needs to be included.

2. Lines 57-58: “ST35/RT046, ST3/RT001, and ST3/RT009 were mostly distributed genotypes of strains in children group” - The authors’ previous publication in Scientific Reports (DOI:10.1038/s41598-018-21762-7) states “ST35/RT046, ST3/RT001 and ST3/RT009 were the commonly found in children patients but ST54/RT012 for adults”. Therefore, lines 57-58 in the present manuscript are repeated reporting of the same result and can amount to plagiarism. The authors may refer to this finding as “We have previously reported…” but not present them in the Results section as done here as a novel finding.

3. Lines 206-218 and Figure 1A, B and C: Unless the authors can adequately justify the basis for the selection of 49 isolates from the previously reported 978, the information in line 206 and Figure 1 have been previously published in the authors’ previous publication. The section entitled ‘General information’ can be removed.

4. Line 53: The method used for antibiotic susceptibility testing (Etest) needs to be mentioned here, as there is some variability of MICs reported from different AST methods. The authors need to also address this variability in the manuscript when interpreting CLSI MIC breakpoints.

5. It will be interesting to see if there is an association (or a lack thereof) between the phenotypic resistance detected by Etest AST and the annotated antibiotic resistance genes in different strains. Some potential mechanistic explanations for the detected resistance are needed.

Validity of the findings

1. Lines 338-340: “The CARD annotation results were consistent to phenotypic results, macB, vanRA, vanRG, vanRM, arlR, and efrB were resistant genes referred to macrolide, glycopeptide, and fluoroquinolone”. Did the CARD annotation reveal the presence of resistance genes in the same isolates in which the specific phenotypic resistance was seen? How is this statement valid, as glycopeptide resistance was not detected in this study? A better and more accurate discussion on this point is needed.

2. Lines 364-373: The role of tcdC in repression of PaLoc toxin expression has been subsequently disproven. The authors should read and refer to the publication by Bakker et al, 2012, entitled ‘TcdC Does Not Significantly Repress Toxin Expression in Clostridium difficile 630ΔErm’(https://doi.org/10.1371/journal.pone.0043247) and alter their discussion.

3. Lines 373-375: “Moreover, the polymorphism of tcdB gene was less than that of tcdA, which showed that the nucleotide polymorphism of tcdB gene was more conserv(ed) than tcdA” – The authors should expand on the discussion of this point based on the differential impact of tcdA and tcdB on the pathogenesis of CDI. (E.g. for relevant references - https://doi.org/10.1093/infdis/jis669).

4. Lines 230-232 and Table 1: How you conducted this comparison needs to be explained better. How where the intermediately resistant strains handled in the Chi-square test?

5. “All the strains in this study were divided into clade 1 and clade 4” – This is an interesting finding and suggests that there is a preponderance of C. difficile strains of clades 1 and 4 in Southwest China. However, I wonder if a different picture will be painted if all the previously reported 978 isolates were studied. This again highlights the paramount importance of the authors providing an adequately justifiable reason for selecting these particular 49 strains from 978, to establish the validity of their findings.

6. Lines 278-279: “The characteristics of genome diversity for clade 1 could be found” – Which characteristics could be found?

7. Lines 346-348: “RT014, RT002 and RT018, all of these strains are the most commonly isolated types of RTs from CDI patients (Bauer et al. 2011; Collins et al. 2013; Foster et al. 2014; Lim et al. 2014)” – This statement is not accurate. The studies the authors have referenced have used different methodologies and a blanket comparison between them to declare some as ‘the most commonly isolated types’ (while leaving out other ribotypes highly reported by the same studies) is inaccurate. You may refer to these RTs as ‘some of the most commonly isolated’ or a similar term and highlight that they have been isolated from multiple geographical locations in the world.

Additional comments

Please follow the formatting guidelines described in the ‘Instructions for Authors’ section of the journal.

Reviewer 3 ·

Basic reporting

Clostridium difficile infection (CDI) is a significant clinical problem. The data this study provides and its thorough analysis can be beneficial, insightful, and provides unique information. The effort and analytics put forward by the authors are appreciated. This communication is a step in the right direction towards a complete understanding and will open avenues for further research.
The manuscript figures are nicely designed. The work provides an important addition to the understanding of antibiotic resistance in CDI.

Comments.

1. The manuscript needs to be read through by a fluent English speaker to correct its numerous grammatical and sentencing errors.

Experimental design

Comments:

1. Patient selection criteria should at least be mentioned, I understand the samples are from a previous study but this is a new publication and should give a complete picture by itself.

2. The authors use 49 C. difficile isolates in the current study, but the selection criteria for why these specific 49 isolates were selected from the set of isolates used in the previous study are not mentioned.

Validity of the findings

The background and rationale of the study are relevant, and the data support the conclusions. The significance of this study is appreciated

---

## Round 0.2 · Minor Revisions

There are some minor issues or questions reviewer mentioned, please address them, thanks.

Reviewer 2 ·

Basic reporting

Lines 58-61: Suggestion for language correction “Forty nine strains of C. difficile were used in this study. Five isolates were non-toxigenic and the rest carried toxigenic genes. We have previously reported that ST35/RT046, ST3/RT001 and ST3/RT009 were the mostly distributed genotypes of strains in the children group”

Q3:
Grammatically incorrect. Please re-phrase this.

Q43:
The description of genomic features needs to include some comparison to the previously reported C. difficile strains.
Also, what is meant by ‘effect reads’ is not clear.

Q47:
‘CARD’ refers to ‘Comprehensive Antibiotic “Resistance” Database’, not “Research”.

Lines 387-391: Please improve grammatical accuracy.

Lines 429-431 and 447-448: Rather than repeating the information on tcdB gene by referring to both polymorphisms and conservation, the sentence can be better structed. For example: “The tcdB gene was more conserved than the tcdA gene” or ““The tcdB gene had fewer polymorphisms than the tcdA gene”. Also, please improve grammatical accuracy.

Line 433: It is not needed to start the words tcdB and tcdA with an uppercase “T”.

A80:
Suggested language edit for “The characteristics of highly genomic diversity of isolates for clade 1 could be found” – “The high genomic diversity characteristic of clade 1 could be detected in the strains that were grouped into this clade”.

Experimental design

Q75:
I suggest editing “except the vanRA, vanRG and vanRM for glycopeptide resistance” as “except for vanRA, vanRG and vanRM and glycopeptide resistance, which was not phenotypically tested” to clarify that glycopeptide resistance was not phenotypically tested in this study, rather than no association was detected.

A78:
Thank you for your explanation. Please add this information on the statistical tests used to the footnote of Table 1.

Validity of the findings

No comment.

Additional comments

The edited manuscript has substantially improved with regard to both the science and writing/ presentation. A few minor concerns are listed above.

---

## Round 0.3 · accepted · Accept

You have revised the manuscript according to the reviewer's comment, and now it is accepted, thank you.